# Haemodynamic Forces: Emerging Markers of Ventricular Remodelling in Multiple Myeloma Cardiovascular Baseline Risk Assessment

**DOI:** 10.3390/cancers16173081

**Published:** 2024-09-04

**Authors:** Anna Colomba, Anna Astarita, Giulia Mingrone, Lorenzo Airale, Cinzia Catarinella, Fabrizio Vallelonga, Dario Leone, Marco Cesareo, Arianna Paladino, Sara Bringhen, Francesca Gay, Gianni Pedrizzetti, Franco Veglio, Alberto Milan

**Affiliations:** 1Candiolo Cancer Institute, FPO—IRCCS, 10060 Candiolo, Italy; anna.colomba@unito.it (A.C.); giulia.mingrone@unito.it (G.M.); fabrizio.vallelonga@ircc.it (F.V.); ariannapaladino@hotmail.it (A.P.); alberto.milan@unito.it (A.M.); 2Department of Medical Sciences, University of Turin, 10126 Turin, Italy; 3Hypertension Unit, Department of Medical Sciences, Division of Internal Medicine, AOU Città della Salute e della Scienza University Hospital, 10126 Turin, Italy; anna.astarita@unito.it (A.A.); lorenzo.airale@unito.it (L.A.); cinzia.catarinella@unito.it (C.C.); marcor.cesareo@gmail.com (M.C.); franco.veglio@unito.it (F.V.); 4SSD Clinical Trial in Oncoematologia e Mieloma Multiplo, Division of Hematology, AOU Città della Salute e della Scienza University Hospital, 10126 Turin, Italy; sarabringhen@yahoo.com (S.B.); francesca.gay@unito.it (F.G.); 5Department of Molecular Biotechnology and Health Sciences, Division of Hematology, University of Torino, 10124 Turin, Italy; 6Department of Engineering and Architecture, University of Trieste, 34127 Trieste, Italy; gianni.pedrizzetti@dia.units.it

**Keywords:** multiple myeloma, cardiovascular risk, echocardiography, haemodynamic forces, arterial hypertension

## Abstract

**Simple Summary:**

Multiple myeloma (MM) patients are often affected by cardiovascular (CV) diseases, making baseline CV risk evaluation a fundamental step before starting cardiotoxic drug regimens. Haemodynamic forces (HDFs) analysis is the latest technology for the early identification of myocardial damage. We aimed to identify differences in HDFs analysis in patients with MM, hypertension or both versus normotensive non-oncologic subjects. During echocardiography and pulse wave velocity assessment, hypertensive patients showed decreased ejection fraction, global longitudinal strain and HDFs values compared with normotensive non-oncologic patients, whereas ventricular mass and PWV increased. Multiple myeloma normotensive patients displayed a significant reduction in systolic HDFs and systolic ejection HDFs compared with normotensive non-oncologic patients, but no significant change in terms of standard ventricular markers and PWV was found. Therefore, MM leads to ventricular remodelling regardless of hypertension; HDFs act as early markers of subclinical cardiac damage, and we propose HDFs analysis application in normotensive oncologic patients.

**Abstract:**

Multiple myeloma (MM) affects a population with a high prevalence of cardiovascular (CV) disease. These patients benefit from an accurate CV risk evaluation in order to choose the safest drug regimen. Haemodynamic forces (HDFs) analysis allows for the earlier detection of myocardial damage compared with standard markers; the role played by MM in HDFs alteration, with or without the influence of hypertension, is yet to be studied. Therefore, we aimed to identify differences in HDFs analysis in patients with MM, hypertension or both versus normotensive non-oncologic subjects. A total of 173 patients (MM hypertensive patients, MMHT; MM normotensive patients, MMNT; non-oncologic hypertensive patients, CoHT; and non-oncologic normotensive patients, CoNT) underwent transthoracic echocardiography for HDFs analysis and pulse wave velocity (PWV) assessment. Hypertensive patients (MMHT, CoHT) showed decreased ejection fraction (EF), global longitudinal strain (GLS) and HDFs values compared with CoNT, whereas ventricular mass (LVMi) and PWV increased. MMNT displayed a significant reduction in systolic HDFs (*p* < 0.006) and systolic ejection HDFs (*p* < 0.008) compared with CoNT, without significant change in EF, GLS, LVMi or PWV. In conclusion, MM leads to ventricular remodelling regardless of hypertension; HDFs application for MM patients could help detect early myocardial damage, especially in patients receiving cardiotoxic drugs.

## 1. Introduction

Multiple myeloma (MM) is a plasma cell dyscrasia affecting the older population, with an average age at diagnosis of about 69 years old [1]; this is an age group notoriously impacted by cardiovascular (CV) disease [2,3]. CV comorbidities in MM patients may derive from three different mechanisms [4]. Above all, these include patient-related factors, such as age itself, arterial hypertension, dyslipidemia and other CV risk factors. Secondly, multiple myeloma-related factors: blood hyperviscosity and thrombosis, anaemia, renal failure and AL-amyloidosis, eventually resulting in high-output heart failure [5]. Finally, drug-related factors such as exposure to some anti-myeloma therapeutic regimens, such as proteosome inhibitor Carfilzomib could lead to worsening myocardial or vascular damage, both potential manifestations of cardiotoxicity [6,7]. There is a significant prevalence of CV disease in MM versus non-MM patients [8], one of the most clinically important being arterial hypertension, defined by repeated office blood pressure (BP) values greater than 140 mmHg (systolic) and 90 mmHg (diastolic). [9] At the time of MM diagnosis, patients show a 30% higher incidence of arterial hypertension than comparable non-oncologic populations [10], and arterial hypertension itself is a risk factor for chemotherapy-induced cardiotoxicity [3] and a negative prognostic factor for survival [11]. The baseline CV risk in this population strongly influences the absolute risk of CV complications during or after treatment, apart from exposure to cardiotoxic therapies. Therefore, current evidence recommends the careful assessment of oncologic patients for CV and Cardiovascular Adverse Event (CVAE) risk stratification before starting anti-cancer treatment [12,13]. CV imaging has a central role in risk stratification as it allows us to detect subclinical organ damage, the first-line imaging method being trans-thoracic echocardiography (TTE). Unlike the traditional echocardiographic technique based on volume estimation (ejection fraction, EF), speckle-tracking echocardiography enables the analysis of left ventricular (LV) wall longitudinal (global longitudinal strain, GLS) and circumferential (global circumferential strain, GCS) deformation, providing an earlier detection of ventricular remodelling than EF [14,15,16]. Similarly, non-invasive haemodynamic forces (HDFs) analysis has emerged as a novel technique which, through speckle-tracking echocardiography, allows us to estimate the intraventricular pressure gradients from time-resolved energy exchange in the blood–chamber interaction [17,18]. As the relationship between blood flow and the ventricular wall is so close, blood flow can be estimated by knowledge of the mitral and aortic orifices’ diameters and endocardial tissue movement identified by speckle-tracking TTE [19,20], potentially making HDFs analysis a more widespread and accessible technology in clinical practise. It was demonstrated that this method is highly effective in detecting early changes in LV function: HDFs modify prior to EF or GLS, thus being the most precocious marker of cardiac remodelling [17,20]. To date, HDFs were examined in relation to heart failure and cardiac resynchronization therapy (whose response they were able to properly predict) [21,22], but their role in detecting myocardial damage and ventricular remodelling in oncologic patients has never been explored.

Through this novel technology, we aim to investigate the potential role of HDFs in detecting ventricular remodelling in MM patients with or without hypertension; to determine their effective advantage and precociousness compared with standard markers of myocardial damage; and to identify the most suitable subgroup of oncologic patients most likely to benefit from HDFs analysis application.

## 2. Materials and Methods

This single-centre cross-sectional study took place at the Hypertension Unit and Centre for Cardiovascular diseases of ‘Città della Salute e della Scienza’ Hospital in Turin, Italy, and was approved by the local bioethics committee of the “A.O.U. Città della Salute e della Scienza” hospital of Turin, Italy (protocol number 0038655). All participants were >18 years of age and had signed an informed consent form in accordance with Declaration of Helsinki. Two groups of patients meeting these inclusion criteria were recruited after clinical evaluation.

The first group included oncologic patients affected by multiple myeloma and were referred to Hypertension Unit by haematologists for CV risk evaluation prior to Carfilzomib-based therapeutic regimens. MM patients were enrolled between April 2017 and November 2023. Those who had been already treated with cardiotoxic drugs and those affected by light-chain cardiac amyloidosis were excluded. 

The second group included hypertensive and normotensive non-oncologic patients medically treated at the Hypertension Unit and enrolled between April 2018 and April 2021, excluding patients with valvular or connective tissue diseases.

### 2.1. Clinical Assessment 

All patients underwent a detailed baseline assessment, in accordance with ESC/ESH guidelines [9,23] and the European Myeloma Network protocol [24], including cardiovascular and oncological history collection, office and ambulatory blood pressure measurement (ABPM), 12-lead electrocardiogram (ECG), arterial stiffness evaluation through pulse wave velocity (PWV) and trans-thoracic echocardiography (TTE) execution. 

Arterial stiffness was estimated measuring PWV via a validated device (Sphygmocor System Atcor Medical, Sydney, Australia). PWV values > 9 m/s were considered indicators of vascular subclinical organ damage [25].

Transthoracic echocardiography was performed by expert EACVI (European Association of Cardiovascular Imaging) accredited staff with an iE33, Affinity 50 or EPIQ7C ultrasound machine (Philips Medical System, Andover, MA, USA) equipped with a sector S5-1 probe for two-dimensional and Doppler acquisition, after placing patients in a left lateral decubitus position. All echocardiographic images analysed were technically of good quality and synchronised with the ECG lead. LV diameters and wall thickness were measured in parasternal long-axis view, deriving LV geometry through the Deveraux formula indexed according to body surface area and height elevated to 2.7 (LVMi). LV hypertrophy (LVH) was considered a marker of cardiac subclinical organ damage from LVMi mass values ≥115 g/m^2^ (≥49 g/m^2.7^) and ≥95 g/m^2^ (≥47 g/m^2.7^) in men and women, respectively. LV ejection fraction (LVEF) was derived from apical 4-chamber and 2-chamber views. Global longitudinal strain (GLS) was studied from apical views using speckle-tracking analysis software (Automated Cardiac Motion Quantification, QLAB Cardiac Analysis ver.15, Philips, Andover, MA, USA), following current recommendations [26].

### 2.2. Haemodynamic Forces

Haemodynamic forces (HDFs) evaluation required the off-line speckle-tracking analysis of echocardiographic DICOM apical 2-, 3- and 4- files after uploading them to dedicated software (QStrain, Medis Medical Imaging Systems 4.0.56.4, Leiden, The Netherlands).

A detailed description of HDFs analysis has been reported in our previous works [27].

Figure 1 displays a typical apical–basal time profile of HDFs. Inside the LV, HDFs occur along 3 planes: apical–basal (longitudinal component), lateral–septal (transversal component) and inferior–anterior. Longitudinal HDFs are described in the literature as predominant under normal conditions [28], and are thus the only ones analysed in the present study. When the apical pressure is higher than the basal, it is conventionally represented on the curve profile as a positive deflection; when the HDFs vector is directed from the base to the apex, this is representative as a negative deflection.

HDFs can be analysed during the entire heartbeat (EH) and during its sub-phases. The systolic phase can be further divided into the following:Acceleration phase (SyAcc_ab), starting at the opening of the aortic valve; the blood pressure gradient is directed from the apex towards the heart base (positive deflection in Figure 1);Ejection phase (SyEj_ab): the blood pressure gradient is directed towards the heart base, but decreases as blood flows through the aortic valve (positive deflection);Deceleration phase (SyDec_ab): as blood enters the arterial system, the pressure vector inside the ventricle inverts (negative deflection).

Diastolic phases are discussed elsewhere [27].

Figure 1 shows the graphical representation of apico-basal ventricular haemodynamic forces. The systolic phase (Sy_ab) is comprehensive of the systolic acceleration (Syacc_ab), ejection (SyEj_ab) and deceleration (SyDec_ab) subphases. Systolic acceleration and deceleration are indicated by pointers. Systolic ejection is represented by the horizontal arrow. The only diastolic subphases examined in the present study are the relaxation (DiRelax_ab) and deceleration (DyDec_ab) subphases, indicated by pointers. Further diastolic phases are not explored nor represented.

### 2.3. Statistical Analysis

Statistical analysis was performed using dedicated software (IBM SPSS Statistics, Version 25.0.0.0, IBM Corp., Armonk, NY, USA; Jamovi, version 2.2.5). The data were presented as “mean ± standard deviation” or “median [inter-quartile range]” or “observations (percentage frequency)” as appropriate. The Student’s *t*-test or Wilcoxon test for quantitative variables, the McNemar test for qualitative variables, and the One-Way ANOVA test were used to analyse differences between the groups. A *p* < 0.050 for two-tailed tests was considered significant in all statistical analyses.

## 3. Results

Out of 112 patients with MM, 104 hypertensive patients and 120 normotensive patients evaluated at our centre, respectively, 71, 52 and 50 were included in the following analyses (see the flowchart of the study population in Appendix A).

### 3.1. General Characteristics and Cardiovascular Risk Factors

Patients with MM had a median age of 69 [11] years. The MM subtype was relapsed–refractory in the majority of cases (67 out of 71), with only four patients being newly diagnosed. Subclinical hypertension-mediated organ damage (HMOD) was present both as left ventricular hypertrophy at echocardiographic evaluation (16.2% of oncologic population) and as increased arterial stiffness (29.7%) (see Table 1).

Non-oncologic hypertensive and normotensive patients had a median age of 60 [26.80] years old, with a statistically significative difference in terms of BMI compared with MM patients (24.80 [5.24], *p* < 0.001). BP values and HMOD prevalence were similar to those of cancer patients (see Table 1).

Table 1 includes the population descriptives. The data are represented as all subjects, MM and non-MM populations. The *p*-value column refers to the comparison between MM and non-MM patients. Office blood pressure, pulse wave velocity and left ventricular mass values are represented. Increased PWV and LVH stand for subclinical organ damage.

Both groups were further separated according to hypertensive status into MM hypertensive patients (MMHT, n = 44), MM normotensive patients (MMNT, n = 27), non-oncologic hypertensive patients (CoHT, n = 52) and non-oncologic normotensive patients (CoNT, n = 50) (see flowchart of the study population in Appendix A). The general characteristics of all the subgroups are listed in Table 2.

Table 2 includes the four groups’ descriptions. Significant comparisons against each group are marked. The *p*-value column refers to the highest *p*-value between groups. Office blood pressure, pulse wave velocity and left ventricular mass values are represented. Increased PWV and LVH stand for subclinical organ damage.

Subclinical organ damage was prevalent in hypertensive groups as HMOD: 36,4% of MMHT and 42.3% of CoHT had LVMi values compatible with a diagnosis of left ventricular hypertrophy; 20.5% of MMHT and 30.8% of CoHT had increased values of PWV, showing signs of arterial stiffness.

### 3.2. Cardiac Deformation Analysis

MM patients (MMHT, MMNT) and non-oncologic groups (CoHT, CoNT) were subsequently studied with a comparative analysis in terms of standard echocardiographic markers (EF, GLS, GCS), haemodynamic forces (HDFs), pulse wave velocity (PWV) and left ventricular mass (LVMi) (see Table 3).

#### 3.2.1. Standard Echocardiographic Markers

Ejection fraction values and strain values, for both global circumferential strain (GCS) and global longitudinal strain (GLS), were significantly reduced in hypertensive patients (MMHT and CoHT) compared with non-oncologic normotensive patients (CoNT, *p* < 0.005, see Table 3).

#### 3.2.2. Haemodynamic Forces Analysis

Apical–basal systolic HDFs, both during the entire systolic period (Sys_ab) and in its sub-phases (SyAcc_ab, SyEj_ab), were significantly reduced in hypertensive patients (MMHT, Sys_ab 12.2%, CoHT, Sys_ab 13.5%) compared with non-oncologic normotensive patients (CoNT, Sys_ab 17.6%, *p* < 0.008, see Table 3).

Haemodynamic forces during entire systole (Sys_ab) and the systolic ejection sub-phase (SyEj_ab) were also significantly lower in oncologic normotensive patients (MMNT, Sy_ab 14%, SyEj_ab 14.1%) than non-oncologic normotensive patients (CoNT, Sy_ab 17.6%, *p* < 0.006, SyEj_ab 17.8%, *p* < 0.008,) (Figure 2, Appendix A).

### 3.3. Subclinical Organ Damage Markers

Pulse wave velocity (PWV) and left ventricular mass (LVMi) values were significantly increased in hypertensive patients, both oncologic and non-oncologic (MMHT and CoHT) compared with normotensive patients (MMNT, CoNT). There was no difference in terms of subclinical organ damage markers between normotensive populations (MMNT compared with CoNT) and hypertensive groups (MMHT compared with CoHT, see Table 3).

Table 3 includes the four groups’ descriptions. Significant comparisons against each group are marked. The *p*-value column refers to the highest *p*-value between groups. Standard echocardiographic markers and haemodynamic force values are represented. Increased PWV and LVMi stand for subclinical organ damage.

Figure 2 is a graphical representation of the comparative analysis between the groups. The four box plots include the mean value for systolic ejection haemodynamic force in the respective sub-population. *p*-values < 0.001 are represented in the upper part of the picture.

## 4. Discussion

In this comparative study, we explored the potential utility of HDFs analysis as a precocious marker of subclinical ventricular remodelling. To the best of our knowledge, this is the first work applying this technology to MM normotensive patients, demonstrating its potential utility during baseline CV risk assessment prior to starting cardiotoxic drug regimens.

Haemodynamic forces reflect the changes in the blood–wall relationship [20]. Mechanical forces play an active role during morphogenesis, as intracardiac stress imparted by blood flow influences the macroscopic shaping of the heart [29]; later in life, the same mechanism, in the presence of pathological triggers such as arterial hypertension, results in ventricular remodelling. Haemodynamic force alteration is described to be the first identifiable event anticipating this remodelling; therefore, its analysis could provide essential details on subclinical organ damage and increased risk of manifest CV disease.

Given the impact of arterial hypertension in MM patients [10], we separated them into two sub-groups according to blood pressure status: MM hypertensive patients (MMHT) and MM normotensive patients (MMNT). Two non-oncologic hypertensive and normotensive populations (CoHT, CoNT) enrolled in our Hypertension centre were used for comparative analysis.

Hypertensive patients exhibited significant alterations in HDF values. Patients from both hypertensive groups (MMHT, CoHT) showed reduced apical–basal HDFs during the entire heartbeat (EH_ab) and during the systolic (Sys_ab) phase when compared with normotensive non-oncologic patients. HDFs in systolic sub-phases (SyAcc_ab, SyEj_ab) reflect the same results. These results highlight the central role played by hypertension in influencing the relationship between the blood and the ventricular wall, resulting in HDFs’ significant reduction during the systolic phases. All of the standard cardiac markers analysed (EF, GLS and GCS, LVMi), as well as PWV, changed significantly in hypertensive patients from both populations. As expected from precedent findings [14], hypertensive patients show impacted myocardial contractility (with reduced EF and GLS) and signs of subclinical cardiac and vascular organ damage (increased LVMi and PWV). Ultimately, from our results, hypertensive status may be strongly associated with myocardial damage and change in the blood–wall relationship, independently from oncologic status.

In order to understand whether the oncologic disease itself could induce ventricular remodelling, we compared MM normotensive patients with a non-oncologic normotensive population. Our results show a meaningful reduction in terms of HDFs during both entire systole (Sys_ab) and the systolic ejection phase (SyEj_ab) in oncologic normotensive patients compared with non-oncologic normotensive patients. On the other hand, standard echocardiographic markers such as EF, GLS and LVMi, as well as PWV, did not change significantly between the two groups. This finding, consistent with previous works [21,30], demonstrates how blood flow analysis might be able to detect subtle changes in cardiac function more precociously than standard echocardiographic markers, which remain unvaried. Additionally, these results reveal that MM itself may contribute to ventricular remodelling, likely through a multifactorial pathogenesis (blood hyperviscosity, anaemia, and cardiotoxic therapies). For this reason, baseline CV evaluation in MM patients is crucially important.

There are several tools designed for cardiovascular risk stratification in oncologic patients [31,32], with the most notable example being the HFA-ICOS score [33], as well as some management protocol dedicated to MM patients specifically. A more recent work from our group suggested a risk score for CVAEs prediction during treatment with Carfilzomib in refractory-relapsed MM patients involving office systolic blood pressure value, blood pressure variability value at ambulatory blood pressure monitoring (ABPM), GLS value, presence/absence of LVH at echocardiography and PWV value at arterial stiffness estimation [34]. In the present study, we propose haemodynamic forces assessment as part of our routinary baseline evaluation for MM patients, without the need for further imaging acquisition.

Blood flow analysis is still considered an emerging technology, as, to date, only limited evidence is available. Speckle-tracking echocardiography has greatly widened its application, as it is more accessible than 4D flow cardiac magnetic resonance. Our study offers preliminary evidence that HDFs are more precocious and sensitive markers of ventricular remodelling when applied to a specific subgroup of patients for which no standardised management protocol is available. Given this promising capability, blood flow analysis might add useful information to future standard cardiovascular risk assessment protocols for normotensive MM patient candidates for cardiotoxic therapies.

### Study Limitations

Our study has some limitations. First of all, the cross-sectional nature of this study only allows for the observation of associations at a single point in time, limiting the ability to infer causality between MM, hypertension and changes in cardiovascular parameters, implying the use of ad hoc designed longitudinal studies. These longitudinal studies would also help to reduce time bias by shortening the enrolment period, while simultaneously allowing for the monitoring of changes in HDFs over time. The absence of a control group matched to MM patients may have limited the scientific evidence of the results. In most cases, patients were excluded from recruitment because of insufficient quality of echocardiographic images, greatly impacting on sample size. The initial inclusion criteria limit the external validity of the results for the entire MM population, particularly for those with concomitant amyloidosis or a history of cardiotoxic treatment; moreover, the sample size did not permit a thorough analysis of haematological data, such as ISS stage and cytogenetic risk. Because of these limitations, further studies are needed to deepen our knowledge about HDFs in oncologic patients.

## 5. Conclusions

Multiple myeloma is associated with cardiac remodelling. Patients with MM should therefore undergo an accurate cardiovascular risk stratification before starting cardiotoxic drug treatment. In normotensive patients, blood flow analysis is able to detect subclinical myocardial damage before there is a substantial alteration of standard echocardiographic markers and ventricular mass; its integration into management protocols for baseline risk assessment may add useful information. Future studies are necessary to validate and integrate HDF analysis into clinical practise, as well as exploring its applicability to other oncologic populations beside MM patients.

## Figures and Tables

**Figure 1 cancers-16-03081-f001:**
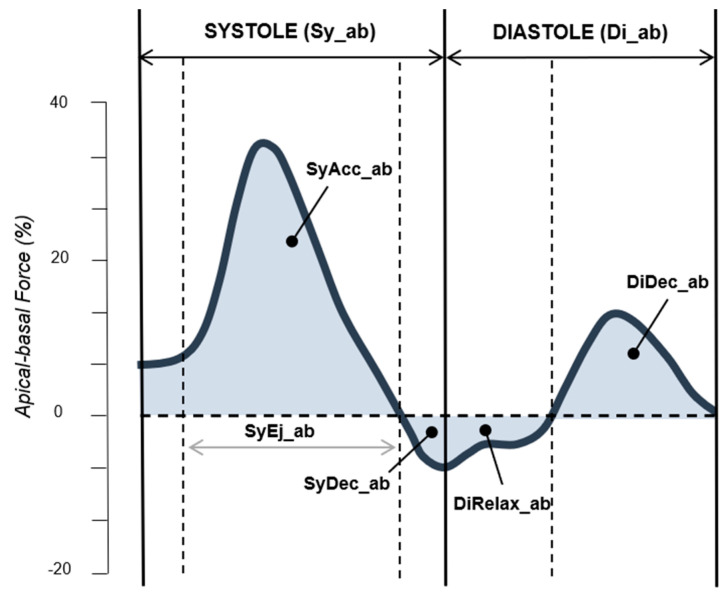
Haemodynamic apical–basal forces pattern. Sy_ab: HDFs during entire systole; Di_ab: HDFs during entire diastole; SyAcc_ab: HDFs during systolic acceleration phase; SyEj_ab: HDFs during systolic ejection phase; SyDec_ab: HDFs during systolic deceleration phase; DiRelax_ab: HDFs during diastolic relaxation phase; DiDec_ab: HDFs during diastolic deceleration phase.

**Figure 2 cancers-16-03081-f002:**
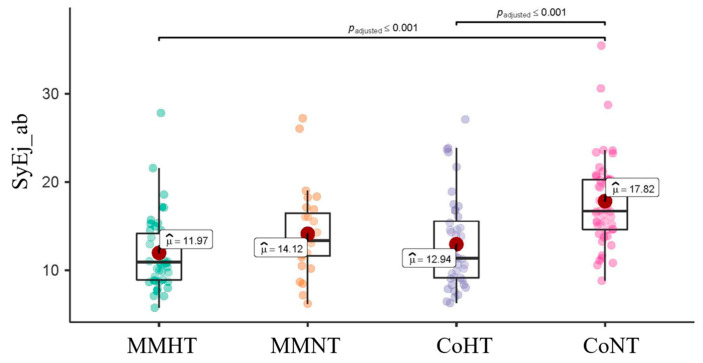
Comparison between groups: systolic ejection subphase. MMHT: multiple myeloma hypertensive group, MMNT: multiple myeloma normotensive group, CoHT: non-oncologic hypertensive group, CoNT: non-oncologic normotensive group, SyEj_ab: HDFs during the systolic ejection phase.

**Table 1 cancers-16-03081-t001:** All subjects’, multiple myeloma and non-MM patients’, general characteristics and subclinical organ damage.

General Characteristics	All Subjects (*n* = 173)	MM Patients (*n* = 71)	Non-MM Patients (*n* = 102)	*p*-Value (MM vs. Non-MM Patients)
Age, years	63 [17]	69 [11]	60 [26.80]	<0.001
Male sex, n (%)	107 (61.80)	45 (63.38)	62 (60.78)	0.73
BMI, kg/m^2^	25.70 [6.02]	27.70 ± 4.08	24.80 [5.24]	<0.001
**Office BP**				
Systolic BP, mmHg	129 ± 18.30	129 ± 17.80	128 ± 18.70	0.67
Diastolic BP, mmHg	75.30 ± 10.20	76.80 ± 9.91	73 [13]	0.11
**Arterial stiffness**				
PWV, m/s	7.90 [2.60]	7.55 [2.63]	8.10 [2.50]	0.16
Increased arterial stiffness (PWV > 9 m/s), n (%)	51 (31.30)	19 (26.76)	32 (31.37)	0.72
**Ventricular Mass**				
LVMi, g/m^2^	81.60 [33.50]	89.10 ± 21.10	75.90 [37]	0.14
LVH (M mass ≥ 115 g/m^2^; F ≥ 95 g/m^2^), n (%)	29 (17.40)	11 (15.49)	18 (17.64)	0.73

MM: multiple myeloma, BMI: body mass index, BP: blood pressure, PWV: pulse wave velocity, M: male, F: female, LVMi: left ventricular mass index, LVH: left ventricular hypertrophy, M: male, F: female.

**Table 2 cancers-16-03081-t002:** General characteristics of MMHT, MMNT, CoHT and CoNT.

General Characteristics	MMHT, *n* = 44	MMNT, *n* = 27	CoHT, *n* = 52	CoNT, *n* = 50	*p*-Value
Age, years	69 [7] ^d^	64.40 ± 10 ^d^	65 [12.50] ^d^	45 [37] ^b c a^	<0.001
Male sex, n (%)	31 (70.50)	14 (51.90)	40 (75.50)	23 (46)	0.009
BMI, kg/m^2^	28.80 ± 4.35 ^d b^	26 ± 3.88 ^d a^	27.10 [4.97] ^d^	22.10 [3.52] ^b a c^	<0.050
**Office BP**					
Systolic blood pressure, mmHg	138 ± 15.90 ^d b^	116 ± 11.10 ^a c^	141 ± 16 ^d b^	115 ± 10.80 ^a c^	<0.001
Diastolic blood pressure, mmHg	80.90 ± 9.35 ^d b^	70.10 ± 9.18 ^a c^	77.80 ± 10.80 ^d b^	68 [9.75] ^a c^	<0.001
**Arterial stiffness**					
Pulse wave velocity, m/s	8.50 ± 1.70 ^b^	7.10 [1.50] ^c a^	9.30 [2.48] ^d b^	7.60 [2.20] ^c^	<0.001
Increased arterial stiffness (PWV > 9 m/s), n (%)	16 (36.40)	3 (11.10) ^c^	22 (42.30) ^d b^	10 (20) ^c^	<0.050
**Ventricular Mass**					
LVMi, g/m^2^	96.90 ± 18.60 ^d b^	77.80 [22.30] ^c a^	98.40 [30.80] ^d b^	67.30 [19.20] ^a c^	<0.001
LVH (M mass ≥ 115 g/m^2^; F ≥ 95 g/m^2^), n (%)	9 (20.50)	2 (7.40)	16 (30.80) ^d^	2 (4) ^c^	<0.001

^a^ *p*-value < 0.050 vs. MMHT. ^b^ *p*-value < 0.050 vs. MMNT. ^c^ *p*-value < 0.050 vs. CoHT. ^d^ *p*-value < 0.050 vs. CoNT. MM: multiple myeloma, MMHT: hypertensive MM patients, MMNT: normotensive MM patients, CoHT: hypertensive non-oncologic patients, CoNT: normotensive non-oncologic patients, BMI: body mass index, BP: blood pressure, LVMi: left ventricular mass index, LVH: left ventricular hypertrophy, PWV: pulse wave velocity, M: male, F: female.

**Table 3 cancers-16-03081-t003:** Comparative study of oncologic hypertensive and normotensive patients with non-oncologic hypertensive and normotensive populations.

	MMHT, *n* = 44	MMNT, *n* = 27	CoHT, *n* = 52	CoNT, *n* = 50	*p*-Value
EF	58.30 ± 5.40 ^d^	60.90 ± 4.10 ^c^	58 ± 4.90 ^d b^	62.70 ± 3.60 ^a c^	<0.005
EndoGCS	−28.70 ± 4.40 ^d^	−30.10 ± 3.10 ^c^	−27.90 ± 3.70 ^d b^	−31.70 ± 3 ^c a^	<0.005
EndoGLS	−21 ± 2.60 ^d^	−22.30 ± 2.60	−20.90 ± 2.50 ^d^	−23.40 ± 2.50 ^a c^	<0.001
EH_ab	9.35 ± 2.62 ^d^	10.90 ± 3.50	9.70 ± 3.30 ^d^	12.70 ± 3.30 ^a c^	<0.001
Sy_ab	12.20 ± 3.50 ^d^	14 ± 4.90 ^d^	13.50 ± 5.40 ^d^	17.60 ± 5.20 ^a c b^	<0.006
SyAcc_ab	11.40 ± 3.70 ^d^	13.10 ± 4.30	12.20 ± 5.30 ^d^	15.60 ± 4.20 ^a c^	<0.001
SyEj_ab	11.97 ± 4.10 ^d^	14.12 ± 4.90 ^d^	12.94 ± 5.20 ^d^	17.82 ± 5.10 ^a c b^	<0.008
SyDec_ab	−7.40 ± 2.20	−7.62 ± 2.20	−6.80 ± 2.20	−7.50 ± 2.80	>0.050
Di_ab	6.90 ± 2.40	7.60 ± 2.70 ^c^	6.20 ± 1.80 ^d b^	7.50 ± 2.30 ^c^	<0.040
DiRelax_ab	−5.90 ± 2 ^c^	−6.90 ± 2.30 ^c^	−4.60 ± 1.60 ^d b a^	−6.90 ± 1.90 ^c^	<0.030
DiDec_ab	4.80 ± 2.50 ^d b^	7.50 ± 3.40 ^c a^	4.90 ± 1.90 ^d b^	8.20 ± 3.50 ^a c^	<0.003
PWV (m/s)	8.50 ± 1.70 ^b^	7.10 [1.50] ^c a^	9.30 [2.48] ^d b^	7.60 [2.20] ^c^	<0.001
LVMi (g/m^2^)	96.90 ± 18.60 ^d b^	77.80 [22.30] ^c a^	98.40 [30.80] ^d b^	67.30 [19.20] ^a c^	<0.001

^a^ *p*-value < 0.050 vs. MMHT. ^b^ *p*-value < 0.050 vs. MMNT. ^c^ *p*-value < 0.050 vs. CoHT. ^d^ *p*-value < 0.050 vs. CoNT. EF: ejection fraction, GCS: global circumferential strain, GLS: global longitudinal strain, Ab: apico-basal, EH_ab: entire heartbeat, Sy_ab: systolic, SyAcc: systolic acceleration, SyE_abj: systolic ejection, SyDec_ab: systolic deceleration, Di_ab: diastole, DiRelax_ab: diastolic relaxation, DiDec_ab: diastolic deceleration, LVMi: left ventricular mass index, PWV: pulse wave velocity.

## Data Availability

The data presented in this study are available in this paper.

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
