# Peer review of "Haemodynamic Forces: Emerging Markers of Ventricular Remodelling in Multiple Myeloma Cardiovascular Baseline Risk Assessment"

_cancers, 2024, doi:10.3390/cancers16173081_

Round 1

Reviewer 1 Report

Comments and Suggestions for Authors

The authors examined the impact of Multiple Myeloma (MM) on cardiovascular health, particularly in relation to hemodynamic forces (HDFs), which are crucial for early detection of myocardial damage. Given that MM often occurs in individuals with a high prevalence of cardiovascular disease, an accurate cardiovascular risk assessment is vital for selecting safe drug regimens. The study aimed to identify differences in HDFs among patients with MM, with or without hypertension, compared to normotensive non-oncologic individuals.

The study involved 173 patients categorized into four groups: MM hypertensive (MMHT), MM normotensive (MMNT), non-oncologic hypertensive (CoHT), and non-oncologic normotensive (CoNT). Transthoracic echocardiography was used to analyze HDFs and assess Pulse Wave Velocity (PWV). The results showed that hypertensive patients (MMHT, CoHT) had decreased ejection fraction (EF), global longitudinal strain (GLS), and HDFs, along with increased left ventricular mass index (LVMi) and PWV, compared to CoNT. Interestingly, MMNT patients exhibited significant reductions in systolic HDFs, despite no significant changes in EF, GLS, LVMi, or PWV.

The authors concluded that MM causes ventricular remodeling independent of hypertension, and the use of HDF analysis in MM patients may help in the early detection of myocardial damage, particularly for those undergoing cardiotoxic treatments.

Limitations

  1. Single-Center Study: The study was conducted at a single center, which may limit the generalizability of the findings. Data from a broader population across multiple centers could provide more robust and universally applicable results.

  2. Cross-Sectional Design: The study's cross-sectional nature only allows for the observation of associations at a single point in time. This limits the ability to infer causality between MM, hypertension, and changes in cardiovascular parameters.

  3. Exclusion Criteria: The exclusion of patients already treated with cardiotoxic drugs and those with light-chain cardiac amyloidosis may result in a sample that does not fully represent the diversity of the MM patient population. This could limit the applicability of the findings to all MM patients.

  4. Limited Analysis of Hemodynamic Forces (HDFs): The study focused on longitudinal HDFs, despite the fact that other components (like lateral-septal and inferior-anterior HDFs) may also be relevant. This narrow focus may overlook important aspects of cardiac function and myocardial damage.

  5. Restricted Time Frame for Non-Oncologic Patients: The non-oncologic patients were enrolled between April 2018 and April 2021, whereas MM patients were enrolled over a much longer period (January 2015 to November 2023). This discrepancy could introduce temporal bias, affecting the comparability of the two groups.

  6. Lack of Longitudinal Data: Without follow-up, the study cannot track the progression of cardiovascular changes over time in relation to MM and its treatment, particularly the long-term effects of Carfilzomib-based regimens.

Suggestions for Improvement

  1. Expand to Multi-Center Studies: Conducting similar studies across multiple centers could enhance the generalizability of the results and provide a more comprehensive understanding of the cardiovascular impact of MM.

  2. Longitudinal Study Design: Future research should consider a longitudinal approach to track changes in cardiovascular health over time, particularly in response to treatment regimens in MM patients.

  3. Include a Broader Patient Population: To better represent the MM population, future studies should consider including patients with varying treatment histories, including those exposed to cardiotoxic drugs and those with cardiac amyloidosis.

  4. Comprehensive HDF Analysis: Future studies should explore all components of HDFs (including lateral-septal and inferior-anterior vectors) to provide a more complete assessment of cardiac function in MM patients.

  5. Extended Follow-Up: Implementing a longer follow-up period would allow researchers to observe the long-term cardiovascular effects of MM and its treatments, offering more insights into the progression of myocardial damage.

  6. Standardize Enrollment Periods: To minimize temporal bias, future studies should standardize the enrollment periods for all patient groups or adjust for this in the analysis.

  7. Here are the key limitations highlighted in this discussion:

    • Absence of a control group perfectly matched to MM patients which could have limited conclusions about HDFs differences between groups.

    • Small sample size due to exclusion of patients with insufficient quality echocardiographic images, impacting statistical power.

    • Limited evidence currently available on blood flow analysis/HDFs in oncologic populations since this is an emerging application of speckle-tracking echocardiography.

    • Study was conducted at a single center, so findings need replication in larger multi-center trials.

    • Mechanistic insights are limited since this was an observational study; cause-effect relationships cannot be proven.

    • Impact of HDFs alterations on long-term clinical outcomes was not assessed.

    • Potential confounding from differences in cardiovascular risk factors/comorbidities between groups cannot be ruled out.

    • Generalizability may be restricted by characteristics of the selected study population.

    In conclusion, while results are promising, larger prospective studies are still needed to validate the clinical utility and prognostic value of HDFs analysis for risk stratification in MM.

  8. If not in the scope of the manuscript highlight the limitations. Finally, 
  9. Release of pro-angiogenic/pro-inflammatory mediators from the remodeled niche may promote subclinical cardiac dysfunction over time. 

  10. Proliferation and migration of endothelial cells regulated by these factors may remodel BM vasculature, impacting intracardiac stress/blood-wall relationship.

  11. Interactions between myeloma cells, endothelial cells, immune cells mediated by adhesion molecules like CD38/CD31 could affect BM blood rheology.(please refer to 

  12. PMID: 31936715 and expand)
Comments on the Quality of English Language

fine

Reviewer 2 Report

Comments and Suggestions for Authors

The original article “Hemodynamic Forces: Emerging Markers of Ventricular Remodelling in Multiple Myeloma Cardiovascular Baseline Risk Assessment.” reported that the difference of cardiac function among myeloma and non-oncologic populations, and the potential affect of hypertension for cardiac function. This study was very interesting and the results of that was reasonable. However, the background pf patients were quite different as the authors described in the study limitation part. I considered that the unified background was a key to do good retrospective study. Additionally, there were several issues in this article.

1.       The backgrounds of patients were quite different, and so the results of this study might not be reproducible.

2.       What are “cardiotoxic drugs”? Additionally, what was the definition of hypertension? Were the patients who treated with antihypertensive agents, such as ACE inhibitor, ARB, and beta blocker, included in this study? The authors should describe the definition of them in detail.

3.       When was the clinical assessment done? Was the time point of them unified among all the study population? The author should describe the time point when clinical assessment was done in detail.

Reviewer 3 Report

Comments and Suggestions for Authors

In this manuscript "Hemodynamic Forces: Emerging Markers of Ventricular Remodelling in multiple Myeloma Cardiovascular Baseline Risk Assessement", a team led by Colomba and Milan, investigate the value of hemodynamic forces  as a early marker to detect the cardiovascular risk. 

Although the author made several interesting points here, there are several weaknesses that preclude to accept on the current version.

1. What is the subtype of multiple myeloma? Does the new diagnosis or relapse in patients have a different effect on the HDF analysis?

2. The author compared the difference between MMHT and coNT to show the remodeling effect of MM. However, the coHT group is supposed to be set as control, instead of coNT.

3. What are the characteristics of these myeloma patients? Will this treatment strategy, ISS stage, and cytogenetic risk affect the HDF analysis? Multiple-way analysis of variance might be needed.

Round 2

Reviewer 2 Report

Comments and Suggestions for Authors

This first revised manuscript was described enough to respond reviewers' comments. I considered this manuscript was suitable to accept for "Cancers".

Reviewer 3 Report

Comments and Suggestions for Authors

The authors’ responses in the revised manuscript mostly addressed those critiques adequately. The resulting manuscript is much improved and can be accepted on the current version.